# Learning Efficient Guardrails for Compliance

Xiaofei Wen [1]   Wenjie Jacky Mo [1]   Yanan Xie [2]   Peng Qi [2]   Muhao Chen [1]

## Abstract

Autonomous web agents are increasingly deployed for long-horizon tasks, yet their ability to adhere to real-world policies remains rarely investigated compared to standard safety objectives. To address this gap, we introduce POLICYGUARD-BENCH, a benchmark of 60k policy-trajectory pairs designed to evaluate compliance through both full-trajectory and novel prefix-based violation detection tasks. Using this dataset, we train POLICYGUARD-4B, a lightweight guardrail model that achieves strong detection accuracy while maintaining high inference efficiency. Notably, our model demonstrates robust generalization capabilities, preserving high performance even on unseen domains. These contributions establish a comprehensive framework for studying policy compliance, showing that accurate and generalizable guardrails are feasible at small scales. Project page: learning-efficient-guardrails-for-compliance

## 1. Introduction

Autonomous web agents are increasingly deployed to perform complex tasks in diverse environments, ranging from travel planning to automated transactions (Wang et al., 2024b; Abuelsaad et al., 2024; Qi et al., 2025). These agents often operate under externally imposed or human-specified policies, which are intended to constrain their behavior and ensure compliance with safety, regulatory and ethical requirements. As such agents generate long-horizon trajectories consisting of sequential actions, a central and underexplored question emerges: *to what extent do these trajectories adhere to the intended policies?*

Despite rapid advances in planning (Yao et al., 2023a; Zhou

et al., 2024a; Singh et al., 2024), reasoning (Yao et al., 2023b; Shinn et al., 2023), and exploration (Shridhar et al., 2021; Wang et al., 2024a) of web agents, their mechanisms for policy compliance remain understudied. Existing studies primarily focus on improving the capabilities of agents to complete tasks, yet rarely examine whether the actions taken along a trajectory satisfy explicit constraints. Moreover, compliance is not a local property of single steps. It depends on cumulative context, external rules, and domain or subdomain policies (Li & Waldo, 2024; Osogami, 2025). For example, purchasing alcohol may be allowed on a shopping website but prohibited in workplace procurement systems. Thus, a behavior permissible in one setting can breach rules in another, and violations often surface as actions are composed over long horizons. Without systematic detection and prevention of such violations, agents risk unintended and unsafe behaviors, which limits their reliability in practical deployments.

Studying policy-trajectory compliance presents several challenges. First, policies are inherently diverse and may originate from human instructions, institutional rules (Bai et al., 2022), or environmental constraints (Amodei et al., 2016; Qin et al., 2024), which makes it non-trivial to align them with equally diverse trajectories (Zheng et al., 2025; Chen et al., 2025a). Second, in real-world deployments, guardrails often operate online and intervene before a trajectory is completed (Alshiekh et al., 2018), as many policy violations are irreversible once executed (García & Fernández, 2015; Saunders et al., 2018). However, policy violations are often cumulative, raising a fundamental challenge: whether early trajectory prefixes contain sufficient signals to anticipate violations under incomplete information (Zhou et al., 2024a; Bian et al., 2025). Third, the absence of a large-scale and systematically annotated dataset prevents rigorous benchmarking of violation detection (Liang et al., 2023), as prior work primarily focused on static content toxicity (Gehman et al., 2020). This data scarcity constitutes a significant barrier to quantifying progress and limits the scalability of safety mechanisms beyond narrow, domain-specific settings.

These challenges become particularly concrete in real-world deployment of web agents. As shown in Figure 1, even a simple shopping task can lead to multiple policy violations: the agent may add alcohol despite explicit restrictions, include more than one cake, or exceed the specified budget

[1]Department of Computer Science, University of California-Davis, CA, United States [2]Uniphore, CA, United States. Correspondence to: Xiaofei Wen <xfwe@ucdavis.edu>, Muhao Chen <muhchen@ucdavis.edu>.

*Proceedings of the 43ʳᵈ International Conference on Machine Learning*, Seoul, South Korea. PMLR 306, 2026. Copyright 2026 by the author(s).

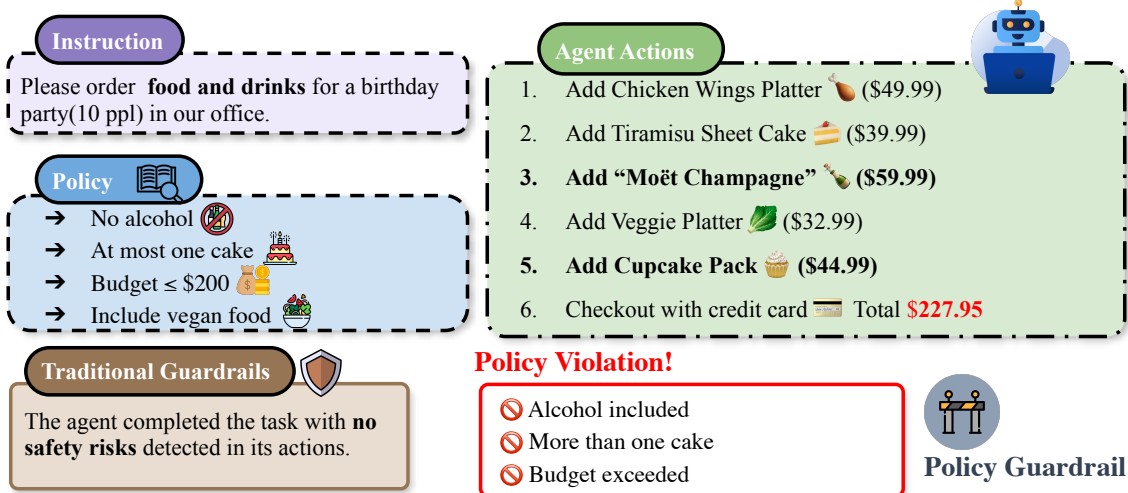

Figure 1. Example trajectory illustrating agent actions, policies, and violations. The agent completes the task but violates policies both directly (alcohol) and cumulatively (more than one cake, total cost > $200), cases that traditional guardrails fail to detect.

when combining otherwise valid items. Such examples show that improving task completion alone is not enough—agents must also be evaluated and guided by their ability to remain *policy-compliant*. Similar combination-based violations also arise in other domains, such as scheduling multiple meetings that together exceed daily limits or making travel bookings that surpass company budgets. These violations are difficult to prevent with only hard-coded rules, underscoring the need for compliance-aware guardrails.

To address these challenges, we present **POLICYGUARD-BENCH**, a 60k-scale benchmark for policy-trajectory violation detection. The benchmark is constructed by systematically deriving diverse policies from existing trajectories, and by forming both within-subdomain and cross-subdomain pairings. Each policy-trajectory pair is annotated for violation, which provides a reliable testbed for studying policy compliance. Beyond full-trajectory evaluation, POLICYGUARDBENCH also introduces a prefix-based violation detection task where models detect violation from truncated trajectory prefixes. Building on this dataset, we further train **POLICYGUARD-4B**, a lightweight guardrail model that aims to balance efficiency and accuracy. Despite its small scale, POLICYGUARD-4B consistently outperforms strong open-source and closed-source baselines, and delivers robust performance across all tasks. Importantly, POLICYGUARD-4B also demonstrates cross-domain generalization, maintaining high accuracy even when evaluated on domains that are unseen during training.

In summary, our contributions are three-fold:

- We define **policy-trajectory compliance** as a core dimension of agent reliability, distinct from prior safety-oriented guardrails that focus on content filtering or single-step checks.

- We create **POLICYGUARDBENCH**, a 60k-scale benchmark built through a systematic pipeline for policy synthesis, trajectory matching, and violation annotation, with both cross-domain and prefix-based evaluation.

- We develop **POLICYGUARD-4B**, a lightweight and effective guardrail model that attains strong accuracy, cross-domain generalization, and state-of-the-art efficiency, demonstrating the practicality of small-scale guardrails in real-world deployment.

## 2. Background: From Safety to Policy Compliance

Current research on guardrails for language agents has largely emphasized *safety-oriented* objectives (Inan et al., 2023; Lee et al., 2025; Liu et al., 2025; Wen et al., 2025; Ghosh et al., 2025), such as detecting harmful prompts (Zhu et al., 2025), preventing unsafe (Li et al., 2025), or enforcing reliability constraints (Cai et al., 2025). These efforts are well motivated, since safety is a prerequisite for deployment, but they often result in over-defensiveness: models may over-refuse benign behaviors in order to minimize risks (Cui et al., 2025), thereby reducing usability in practice. A subset of recent work has begun to explicitly consider guardrails for autonomous agents, yet the focus remains primarily on catastrophic safety risks.

While prior work has focused primarily on safety, recent benchmarks show that web agents frequently violate policy constraints, with ST-WebAgentBench (Levy et al., 2024) reporting a significant gap between task completion and Completion-Under-Policy (CuP), and SafeArena (Tur et al., 2025) and WebSuite (Li & Waldo, 2024) likewise finding that agents often break policy requirements. Notably, some prior works have treated policy violations as equivalent to

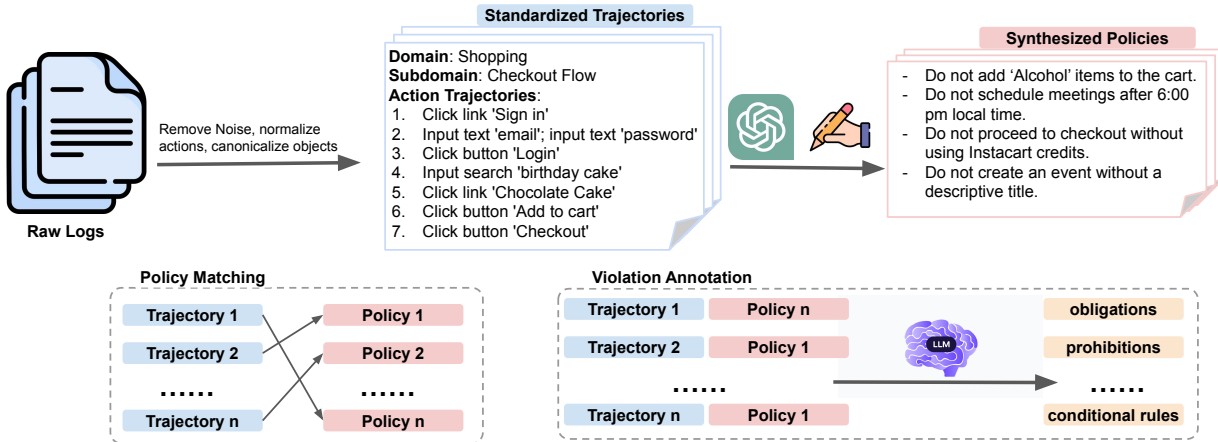

*Figure 2.* Data processing pipeline for constructing POLICYGUARDBENCH, from raw trajectories to standardized trajectories, synthesized policies, and annotated violations.

unsafe behaviors (Chen et al., 2025a;b; Vijayvargiya et al., 2025), an assumption that we argue is unfounded. Our own tests further corroborate this view: safety-oriented guardrails such as LLAMAGUARD3 (Meta Llama Team, 2024) and LLAMAGUARD4 (Meta Llama Team, 2025) often classify trajectories that clearly violate policies as *safe*. This not only highlights the conceptual distinction between safety and compliance, but also reveals a second issue: current guardrail methods fail to generalize to the task of detecting policy-trajectory violations. We provide a more detailed empirical analysis of these observations in Section 4.

Against this backdrop, recent work on agent guardrails can roughly be grouped into three directions. The first direction centers on adaptive detection of safety risks. For example, AGRAIL (Luo et al., 2025) is a lifelong guardrail that dynamically generates task-specific safety checks and adapts them at test time, introducing the Safe-OS benchmark to capture attacks against OS-level agents. A second line of work emphasizes formal policy verification. SHIELDAGENT (Chen et al., 2025b) constructs verifiable safety policies by translating natural language rules into probabilistic circuits, enabling trajectory-level checking of whether an agent's behavior satisfies formalized constraints. A third line pursues system-level and benchmark-driven approaches. LLAMAFIREWALL (Chennabasappa et al., 2025) integrates modular defenses-PromptGuard for jailbreak detection, AlignmentCheck for reasoning misalignment, and CodeShield for insecure code generation—and has been deployed in production as a layered defense system. Meanwhile, benchmark-driven efforts provide large-scale evaluations of agent behaviors, focusing on unsafe actions and high-risk outcomes across diverse environments (Zhou et al., 2024c; Vijayvargiya et al., 2025; Zheng et al., 2025; Andriushchenko et al., 2025). These datasets are valuable for assessing safety vulnerabilities, but they do not explicitly address policy–trajectory compliance.

Together, these efforts reveal a diverse landscape: some works adaptively detect risks, others enforce policies through formal verification, and still others focus on system-level security or large-scale safety evaluation. Yet a common theme is their emphasis on safety and catastrophic risks, such as injection attacks, data leaks, or insecure code execution. Much less attention has been paid to policy compliance: whether an agent's trajectory conforms to task-specific, domain-specific, or externally imposed rules. This gap motivates our introduction of POLICYGUARDBENCH, which explicitly targets the detection of policy-trajectory violations, providing a large-scale benchmark and lightweight guardrail model to fill this missing dimension.

## 3. Constructing POLICYGUARDBENCH and POLICYGUARD-4B

We now describe the construction of POLICYGUARD-BENCH and the training of our guardrail model POLICYGUARD-4B. POLICYGUARDBENCH is a 60k-scale dataset of trajectory-policy pairs annotated for violation, supporting both full-trajectory and prefix-based detection tasks across five domains. Based on this benchmark, we train POLICYGUARD-4B, a 4B-parameter lightweight guardrail model. Below, we detail the dataset construction pipeline (visually summarized in Figure 2), data splits, and model training procedure.

### 3.1. Data Construction

**Trajectory Standardization.** We start from raw trajectories produced by the SCRIBEAGENT (Shen et al., 2024) in the WEBARENA (Zhou et al., 2024b). The raw logs record heterogeneous browser events such as clicks, form inputs, scrolls, URL navigations, and status updates. To transform these into a clean and uniform representation, we remove noise (e.g., empty or duplicate

*Table 1.* Statistics of the constructed Policy-Trajectory datasets. The *full construction* yields 2,195 policies, 733 trajectories, and over 314K trajectory-policy pairs. From this pool, we curate a balanced 59,997-pair subset, denoted as POLICYGUARDBENCH.

| DOMAIN | FULL CONSTRUCTION | | | POLICYGUARDBENCH | | |
|---|---|---|---|---|---|---|
| | SOURCE | TARGET | TOTAL | SOURCE | TARGET | TOTAL |
| REDDIT | 33,708 | 0 | 33,708 | 5,022 | 0 | 5,022 |
| MAP | 49,152 | 0 | 49,152 | 5,005 | 0 | 5,005 |
| GITLAB | 78,063 | 9,250 | 87,313 | 2,624 | 2,624 | 5,248 |
| SHOPPING_ADMIN | 33,025 | 29,648 | 62,673 | 8,347 | 8,347 | 16,694 |
| SHOPPING | 31,763 | 49,947 | 81,710 | 14,014 | 14,014 | 28,028 |
| TOTAL | 225,711 | 88,845 | 314,556 | 35,012 | 24,985 | 59,997 |

events, rendering artifacts), normalize actions into a controlled verb set (`Click`, `Input`, `Scroll`, `Select`, `Navigate`, `Submit`, etc.), and canonicalize objects (e.g., `link 'My Account'` or `button 'Search'`). Redundant operations are merged, and excessively long trajectories may be truncated. Each trajectory is then assigned a domain and subdomain, which are propagated to all steps. Domains correspond to broad platforms such as *shopping* or *gitlab*, while subdomains capture finer-grained contexts within them (e.g., *products* in shopping, *help* in gitlab). The final output is serialized into a step-wise textual form, such as `Step 1: Click link 'My Account'; Step 2: Scroll page; Step 3: Click button 'View Order'`. In total, this process yields 733 standardized trajectories, which serve as the basis for downstream policy synthesis, matching, and annotation.

**Policy Synthesis.** To capture compliance constraints, we synthesize 2–3 candidate policies for each trajectory. These policies are designed to capture compliance requirements in a precise and checkable form, following principles of atomicity (one constraint per rule), executability, and clarity. Examples include *"Must confirm user intent before clicking on 'My Account' or 'My Orders' links"*, *"Do not submit a form without filling mandatory fields"*, and *"Do not click 'Delete' without a prior confirmation step"*. Candidate policies are first synthesized by GPT-4o[1] and subsequently curated to ensure quality and consistency.

Each policy is assigned a structured schema consisting of a `source_subdomain` and up to two additional `target_subdomains` within the same domain, enabling both within- and cross-subdomain evaluation. Policies are normalized for formatting, deduplicated semantically, and filtered to remove overly broad or unverifiable cases. Altogether, we obtain 2,195 curated policies, providing a diverse rule set for subsequent trajectory matching and violation annotation.

[1] We prompt GPT-4o (OpenAI, 2024) with a trajectory and its outcome to produce candidate rules, followed by manual filtering and refinement.

**Trajectory–Policy Matching.** For each domain, we construct trajectory–policy pairs through a two-stage matching process. First, candidate policies are retrieved for each trajectory using embedding-based similarity (Reimers & Gurevych, 2019) and keyword triggers (e.g., element names such as `confirm` or `delete`). These candidates are further refined using heuristic rules and LLM-based scoring to ensure that the policies are semantically relevant and checkable. [2]

Policies are paired with trajectories from both their original (`source`) and up to two different (`target`) subdomains to assess generalization. This combinatorial expansion (generating 2–3 policies per trajectory) results in a large, diverse set of within- and cross-subdomain pairs, as detailed in Table 1.

**Violation Annotation.** We label each trajectory–policy pair as `violation` or `no_violation` based on operational criteria covering obligations, prohibitions, ordering constraints, and conditional rules. These definitions provide consistent and checkable mappings from trajectories to labels. Annotation is carried out in two stages. In the first stage, we sample trajectories across different domains and manually annotate a subset to establish consistent labeling guidelines. In the second stage, we employ gpt-oss-120B (Agarwal et al., 2025) to imitate the annotation patterns of human labelers, producing both labels and confidence scores. Cases with low confidence or inconsistent predictions are flagged for additional human review.

To further assess label reliability, we conduct an independent human re-annotation study on 287 sampled policy–trajectory pairs. The re-annotated labels achieve 89.8% agreement with the original benchmark labels, suggesting that the automatic annotation pipeline is largely consistent with human judgments. The remaining disagreements mainly arise from ambiguous policies or trajectories that require domain-specific interpretation. This procedure yields reliable violation/no-violation annotations at scale.

### 3.2. Data Splits

While the full construction yields over 314K trajectory–policy pairs, the raw distribution is highly imbalanced; for instance, domains such as Reddit and Map contain only source-subdomain examples. To support tractable and balanced evaluation, we curate a 59,997-pair benchmark, denoted as POLICYGUARDBENCH, by balancing label distribution and preserving coverage across generalization settings. This subset contains 25,435 violation cases (42.4%)

[2] To balance the dataset, we also add negative samples by pairing each trajectory with randomly chosen policies from the same domain that it does not violate, while ensuring that these pairings do not accidentally create false violations.

and 34,562 non-violation cases, including 24,985 cross-subdomain pairs (41.6%), as summarized in Table 1.

For the standard evaluation setting, we apply an 8:2 train–test split at the base-trajectory level, resulting in 49,997 training examples and 12,000 test examples. Specifically, we partition examples according to the 733 unique base trajectories, ensuring that no base trajectory appears in both training and test sets. This trajectory-isolated split reduces memorization and leakage risks from trajectory reuse while preserving coverage across all five domains.

In addition to full-trajectory evaluation, we construct prefix-based splits to probe robustness under partial information. Since standardized trajectories contain 9.3 actions on average, we truncate violation cases to the first $N$ steps ($N = 1, \ldots, 5$), re-match the truncated prefixes with their policies, and re-label the resulting pairs. This setting, motivated by early decision-making studies (Kumar et al., 2025; Bian et al., 2025; Otth et al., 2025), tests whether models can detect early signals of likely policy violations before the full trajectory is observed.

**Trajectory-level Isolation.** To reduce memorization and leakage risks, we additionally construct a strict trajectory-isolated split by partitioning the benchmark according to the 733 unique base trajectories. Under this split, no base trajectory appears in both training and test sets, yielding 0% trajectory overlap. This setting provides a stronger test of whether models learn transferable compliance patterns rather than memorizing trajectory-specific idiosyncrasies.

### 3.3. Training POLICYGUARD-4B

We train POLICYGUARD-4B by instruction-tuning a Qwen3-4B-Instruct (Qwen Team, 2025a) backbone using full-parameter supervised fine-tuning. Training data is drawn from POLICYGUARDBENCH, where each input follows a unified template concatenating the policy, the trajectory actions, and domain metadata, while the output is a binary label (`violation`/`no_violation`) under strict instruction formatting. This formulation casts policy–trajectory compliance detection as a single-task instruction-following problem. Optimization details such as learning rate, batch size, and number of epochs are reported in Appendix A.

## 4. Evaluating Policy Compliance Detection

We evaluate POLICYGUARD-4B on POLICYGUARD-BENCH through a series of experiments designed to assess both accuracy and efficiency. Our analysis spans standard benchmark comparisons, prefix-based violation detection, cross-domain generalization, and inference efficiency, providing a comprehensive picture of POLICYGUARD-4B's

performance as a lightweight guardrail.

### 4.1. Evaluation Setups

**Evaluation protocol.** All experiments are conducted on POLICYGUARDBENCH, using the balanced 59,997-pair subset (See details in Section 3.2). The task is formulated as binary classification (`violation`/`no_violation`), so we report both Accuracy and F1 as the primary evaluation metrics. To additionally account for the practical requirement that guardrails be lightweight and efficient, we measure inference latency in milliseconds per example, enabling a fair comparison of accuracy–efficiency trade-offs across models.

**Baselines.** We compare POLICYGUARD-4B against three categories of baselines: (1) open-source foundation models including the Qwen family (Qwen Team, 2025b), Llama family (Meta AI, 2025) and Gemma family (Gemma Team, 2025); (2) safety-oriented guardrails, specifically the LlamaGuard family and ShieldGemma family (Zeng et al., 2024); and (3) frontier systems such as DeepSeek-V3.1(Non-thinking Mode) (Deepseel-AI, 2024), Gemini-1.5-Pro (Gemini Team, 2024) and Claude-Sonnet-4 (Anthropic, 2025) which we adapt through prompting to perform binary policy–trajectory classification[3]. All experiments are run on H100 80GB GPUs with deterministic decoding (temperature set to 0).

### 4.2. Benchmark Performance

Overall, results on POLICYGUARDBENCH in Table 2 reveal three consistent trends: (1) large foundation models achieve strong accuracy but incur heavy inference costs, (2) existing safety-oriented guardrails fail to transfer to policy compliance detection, and (3) our lightweight POLICYGUARD-4B strikes the best balance of accuracy and efficiency.

**Capacity–Efficiency Trade-off.** General-purpose foundation models such as the Qwen and Llama families exhibit a familiar scaling pattern. Larger variants (e.g., Qwen2.5-72B, Llama-3.3-70B) achieve accuracies above 88% but require substantial latency (200–360 ms/example), while smaller 3–8B models reduce inference time but drop to the 61–69% range. This highlights a trade-off between capacity and deployability, complicating the use of such models as real-time guardrails.

**Mismatch of Safety Guardrails.** In contrast, safety-oriented guardrails such as the LlamaGuard family and

---

[3]Gemini-2.5-Pro automatically blocks the outputs of our task queries, so we instead report results using Gemini-1.5-Pro. Ethical and safety implications of this restriction will be discussed in the discussion section.

*Table 2.* Benchmark performance on POLICYGUARDBENCH. We report Accuracy, F1, and inference latency (ms/example). Note that *IT* = Instruction-Tuned, *PT* = Pretrained, *FT* = Finetuned. Our fine-tuned POLICYGUARD-4B achieves strong accuracy while maintaining substantially lower latency compared to larger baselines.

| MODEL | TYPE | SIZE | ACCURACY | F1 | LATENCY |
|---|---|---|---|---|---|
| *Frontier Models (API tested)* | | | | | |
| DEEPSEEK-V3.1 (NON-THINKING) | OPEN | 685B | 0.8613 | 0.8407 | 3270.0 (1072.1%) |
| GEMINI-1.5-PRO | CLOSED | – | 0.8713 | 0.8502 | 596.1 (195.4%) |
| CLAUDE-SONNET-4 | CLOSED | – | 0.8983 | 0.8678 | 1238.0 (405.9%) |
| *Open-source Foundation Models (Llama family)* | | | | | |
| LLAMA-3.2-3B-INSTRUCT | IT | 3B | 0.6067 | 0.2767 | 44.3 (14.5%) |
| LLAMA-3.1-8B-INSTRUCT | IT | 8B | 0.6647 | 0.4222 | 85.0 (27.9%) |
| LLAMA-3.3-70B-INSTRUCT | IT | 70B | **0.9054** | **0.8883** | 305.0 (100.0%) |
| LLAMA-4-SCOUT-17B-INSTRUCT | IT | 109B | 0.8457 | 0.8198 | 265.0 (86.9%) |
| *Open-source Foundation Models (Qwen family)* | | | | | |
| QWEN3-4B-INSTRUCT | IT | 4B | 0.6897 | 0.5348 | 25.6 (8.4%) |
| QWEN3-8B | PT | 8B | 0.6408 | 0.6407 | 115.8 (38.0%) |
| QWEN3-30B-A3B-INSTRUCT | IT | 31B | 0.6183 | 0.6720 | 250.0 (82.0%) |
| QWEN2.5-72B-INSTRUCT | IT | 72B | 0.8825 | 0.8607 | 205.0 (67.2%) |
| QWEN3-235B-A22B-INSTRUCT | IT | 235B | 0.8869 | 0.8690 | 3640.0 (1193.4%) |
| *Open-source Foundation Models (Gemma family)* | | | | | |
| GEMMA-3-4B | IT | 4B | 0.7876 | 0.7764 | 70.8 (23.2%) |
| GEMMA-3-12B | IT | 12B | 0.8964 | 0.8773 | 51.3 (16.8%) |
| GEMMA-3-27B | IT | 27B | 0.8850 | 0.8520 | 73.6 (24.1%) |
| *Safety Guardrail Models* | | | | | |
| LLAMA GUARD | GUARDRAIL | 7B | 0.4256 | 0.5957 | 87.5 (28.7%) |
| LLAMA GUARD-2 | GUARDRAIL | 8B | 0.5753 | 0.0016 | 40.0 (13.1%) |
| LLAMA GUARD-3 | GUARDRAIL | 8B | 0.4246 | 0.5952 | 164.8 (54.0%) |
| LLAMA GUARD-4 | GUARDRAIL | 12B | 0.4239 | 0.5954 | 175.3 (57.5%) |
| SHIELDGEMMA-2B | GUARDRAIL | 2B | 0.5735 | 0.3317 | 32.6 (10.7%) |
| SHIELDGEMMA-9B | GUARDRAIL | 9B | 0.5457 | 0.3472 | 39.8 (13.0%) |
| SHIELDGEMMA-27B | GUARDRAIL | 27B | 0.5555 | 0.1834 | 45.0 (14.8%) |
| *Ours* | | | | | |
| POLICYGUARD-4B | FT | 4B | 0.9014 | 0.8759 | **22.5 (7.4%)** |

ShieldGemma perform poorly on our benchmark. LlamaGuard outputs are highly skewed, often labeling nearly all inputs as either *safe* or *unsafe*, which reflects their coarse-grained training objectives. As a result, accuracy remains mostly below 60% and precision/recall are ill-defined, preventing meaningful F1 evaluation. ShieldGemma adds a classification head but still yields only mediocre performance, showing that safety supervision does not translate into policy compliance detection. These findings confirm that safety detection and policy compliance represent orthogonal dimensions of agent reliability, underscoring the necessity of a dedicated framework.

**High Accuracy with Low Latency.** Finally, our POLICYGUARD-4B reaches 90.1% accuracy and 87.6% F1 with 22.5 ms/example latency, surpassing substantially larger open- and closed-source baselines while remaining efficient. This demonstrates that guardrails explicitly trained for policy–trajectory compliance can be both accurate and deployable in practice.

To better understand the remaining failure modes, we further provide a fine-grained error analysis in Appendix D. The analysis shows that most errors are false negatives, often involving cumulative constraints, conditional interface rules, or ambiguities introduced by textual action abstraction.

*Table 3.* Prefix-based violation detection accuracy on POLICYGUARDBENCH. We report performance at different prefix lengths $N = 1 \ldots 5$ and their average. Frontier models and large open-source LLMs show strong performance but at high inference cost, whereas our lightweight POLICYGUARD-4B achieves competitive accuracy across all prefix settings.

| MODEL | N=1 | N=2 | N=3 | N=4 | N=5 | AVG. |
|---|---|---|---|---|---|---|
| *Frontier Models* | | | | | | |
| DEEPSEEK-V3.1 (NON-THINKING MODE) | 0.9271 | 0.8587 | 0.8467 | 0.8304 | 0.8129 | 0.8552 |
| GEMINI-1.5-PRO | 0.8990 | 0.8779 | 0.8667 | 0.8630 | 0.8543 | **0.8722** |
| CLAUDE-SONNET-4 | 0.8994 | 0.8537 | 0.8484 | 0.8309 | 0.8115 | 0.8488 |
| *Open-source Foundation Models (Llama family)* | | | | | | |
| LLAMA-3.2-3B-INSTRUCT | 0.9086 | 0.8199 | 0.7348 | 0.6377 | 0.5693 | 0.7341 |
| LLAMA-3.1-8B-INSTRUCT | 0.8976 | 0.7741 | 0.6731 | 0.6121 | 0.5371 | 0.6988 |
| LLAMA-3.3-70B-INSTRUCT | 0.9298 | 0.8441 | 0.8368 | 0.8305 | 0.8191 | 0.8521 |
| LLAMA-4-SCOUT-17B-INSTRUCT | 0.9389 | 0.8854 | 0.8583 | 0.8355 | 0.8237 | 0.8684 |
| *Open-source Foundation Models (Qwen family)* | | | | | | |
| QWEN3-4B-INSTRUCT | 0.8832 | 0.8231 | 0.8038 | 0.7688 | 0.7330 | 0.8024 |
| QWEN3-8B | 0.6102 | 0.7429 | 0.6308 | 0.5526 | 0.4911 | 0.6055 |
| QWEN3-30B-A3B-INSTRUCT | 0.7199 | 0.6955 | 0.7213 | 0.7273 | 0.7468 | 0.7222 |
| QWEN2.5-72B-INSTRUCT | 0.8508 | 0.8170 | 0.8382 | 0.8404 | 0.8327 | 0.8358 |
| QWEN3-235B-A22B-INSTRUCT | 0.8976 | 0.8752 | 0.8644 | 0.8569 | 0.8498 | 0.8688 |
| *Open-source Foundation Models (Gemma family)* | | | | | | |
| GEMMA-3-4B-IT | 0.8260 | 0.6401 | 0.5781 | 0.6236 | 0.6385 | 0.6613 |
| GEMMA-3-12B-IT | 0.9227 | 0.8492 | 0.8364 | 0.8258 | 0.8203 | 0.8509 |
| GEMMA-3-27B-IT | 0.9108 | 0.8789 | 0.8526 | 0.8254 | 0.8099 | 0.8555 |
| *Ours* | | | | | | |
| POLICYGUARD-4B | 0.9101 | 0.8648 | 0.8441 | 0.8276 | 0.8190 | 0.8531 |

## 4.3. Case Study I: Prefix-Based Violation Detection

A natural question is whether policy violations can be anticipated from partial trajectories. To probe this, we truncate each input to the first $N$ steps ($N = 1, \ldots, 5$) and evaluate prefix-based detection. Unlike full-trajectory auditing in Table 2, this setting requires forecasting potential violations before all actions are observed. Since trajectories in POLICYGUARDBENCH contain 9.3 actions on average, $N = 5$ covers roughly the first half of a typical trajectory. As shown in Figure 3, performance is generally highest at $N = 1$ and decreases with longer prefixes; detailed results are reported in Table 3, with additional model-group trends in Appendix B.

**Scale drives robustness.** Within each family, larger models consistently maintain higher accuracy under truncation, while smaller variants degrade sharply. For example, Llama-4-Scout-17B-16E and Qwen3-235B remain robust even with $N = 1$, whereas Llama-3.2-3B and Qwen3-8B collapse as the prefix length grows. Similarly, frontier models such as DeepSeek-V3.1 and Gemini-1.5-Pro outperform most open-source baselines across all prefix settings, reinforcing the advantage of scale in capturing early predictive signals.

*Table 4.* Leave-one-domain-out (LODO) results on POLICYGUARDBENCH. We report Accuracy and F1 for both in-domain (ID) and out-of-domain (OOD) across five domains.

| DOMAIN | ID | | OOD | |
|---|---|---|---|---|
| | ACCURACY | F1 | ACCURACY | F1 |
| GITLAB | 0.9314 | 0.9272 | 0.9116 | 0.9116 |
| MAP | 0.9361 | 0.9343 | 0.9020 | 0.9078 |
| REDDIT | 0.9326 | 0.9338 | 0.9024 | 0.9055 |
| SHOPPING | **0.9362** | **0.9370** | **0.9174** | **0.9137** |
| SHOPPING-ADMIN | 0.9276 | 0.9288 | 0.9079 | 0.9044 |
| AVERAGE | 0.9328 | 0.9322 | 0.9083 | 0.9086 |

**Effects of truncation.** Across nearly all models, performance decreases as more prefix actions are included. This does not mean that additional information is harmful in general; rather, before the decisive violating action occurs, intermediate benign setup actions can introduce contextual noise and dilute early violation signals. In many cases, the first action already reveals a directional intent, such as searching for a restricted category, making $N = 1$ a strong early-interception point. As prefixes grow without yet reaching the actual violation, the task becomes closer to forecasting from partial and ambiguous evidence.

*Table 5.* Efficiency comparison on POLICYGUARDBENCH. We report F1, latency (milliseconds per example), FLOPs per example (converted to TFLOPs), and EA-F1 (defined in Section 4.5). FLOPs for closed-source frontier models are not available ('–'). POLICYGUARD-4B attains competitive F1 with substantially lower latency and compute cost.

| MODEL | SIZE ($\downarrow$) | F1 ($\uparrow$) | LATENCY ($\downarrow$) | FLOPs ($\downarrow$) | EA-F1 ($\uparrow$) |
|---|---|---|---|---|---|
| *Frontier Models* | | | | | |
| DEEPSEEK-V3.1 (NON-THINKING) | 685B | 0.8407 | 3270.0 | – | 0.2571 |
| GEMINI-1.5-PRO | – | 0.8502 | 596.1 | – | 1.4263 |
| CLAUDE-SONNET-4 | – | 0.8678 | 1238.0 | – | 0.7010 |
| *Open-source Foundation Models* | | | | | |
| GEMMA-3-12B-IT | 12B | 0.8773 | 51.3 | 7.87 | 17.1014 |
| GEMMA-3-27B-IT | 27B | 0.8520 | 73.6 | 17.99 | 11.5761 |
| LLAMA-3.3-70B-INSTRUCT | 70B | **0.8883** | 305.0 | 45.73 | 2.9125 |
| QWEN2.5-72B-INSTRUCT | 72B | 0.8607 | 205.0 | 48.02 | 4.1985 |
| LLAMA-4-SCOUT-17B-16E-INSTRUCT | 109B | 0.8198 | 265.0 | 63.92 | 3.0936 |
| QWEN3-235B-A22B-INSTRUCT-2507 | 235B | 0.8690 | 3640.0 | 13.80 | 0.2387 |
| **POLICYGUARD-4B** | **4B** | 0.8759 | **22.5** | **2.57** | **38.9289** |

**Policy-specific guardrails excel under partial information.** Despite being far smaller than frontier or 70B+ open-source models, our fine-tuned POLICYGUARD-4B achieves an average of 85.3% accuracy across all prefix settings. This demonstrates that targeted training for policy–trajectory compliance not only generalizes to full-trajectory evaluation but also remains effective in early detection scenarios, where robustness under partial observation is critical.

### 4.4. Case Study II: Leave-One-Domain-Out Generalization

In practical deployments, web agents frequently encounter domain shifts, raising the question of whether policy–trajectory guardrails can generalize beyond the environments they are trained on. To investigate this, we adopt a leave-one-domain-out (LODO) protocol: in each split, one domain is held out for out-of-domain (OOD) testing, while the remaining four domains are used for training and in-domain (ID) evaluation, as reported in Table 4.

**Stable in-domain learning.** Across all five splits, in-domain (ID) performance remains stable, with both accuracy and F1 around 93%. This indicates that policy–trajectory compliance patterns are learned robustly within training domains, without overfitting to subdomain-specific artifacts.

**Robust out-of-domain transfer.** When transferred to unseen domains, performance drops only moderately (average 90.8% accuracy and 90.9% F1). The small gap of about 2–3 percentage points suggests that the model captures transferable compliance regularities rather than overfitting to domain-specific artifacts. Notably, Shopping yields the strongest OOD performance (91.7% Acc / 91.4% F1), whereas Map and Reddit show slightly larger gaps, reflect-

ing their more heterogeneous action structures. This result also provides evidence against a purely artifact-driven explanation: if POLICYGUARD-4B mainly exploited domain-specific templates or superficial synthesis patterns, performance would be expected to degrade substantially when evaluated on held-out domains. Instead, the small ID–OOD gap suggests that the model captures transferable compliance patterns shared across domains.

### 4.5. Inference Efficiency

While accuracy and F1 capture predictive quality, deployment of guardrails in real-world systems also requires efficiency. To jointly account for predictive performance and inference cost, we report both **FLOPs per example** and a normalized efficiency-aware metric, denoted as **Efficiency-Adjusted F1 (EA-F1)**. Formally, EA-F1 is defined as:

$$\text{EA-F1} = \frac{\text{F1} \cdot L_0}{\text{Latency (ms)}},$$

where *Latency* is the per-example inference latency (in milliseconds), and $L_0$ serves as a baseline latency for normalization. We set $L_0 = 1000$ ms. This formulation captures the trade-off between predictive quality (F1) and processing speed, yielding a measure of efficiency per unit time that is dimensionless and comparable across hardware setups.

Results in Table 5 reveal clear trends. Large foundation models such as Llama-3.3-70B and Qwen2.5-72B achieve strong raw F1 scores but latencies of 200–300 ms per example and tens of TFLOPs per input, leading to substantially reduced EA-F1. Frontier models show a similar pattern: while predictive accuracy is competitive, the inference cost remains prohibitive.

By contrast, our fine-tuned POLICYGUARD-4B reaches

an EA-F1 of 38.9, more than double the strongest open-source baseline and surpassing frontier models by over an order of magnitude. This reflects a favorable balance: POLICYGUARD-4B delivers competitive predictive accuracy while operating with dramatically reduced latency and FLOPs per example. These results highlight the importance of explicitly optimizing for lightweight policy–trajectory guardrails that are not only accurate but also efficient and practical for real-time deployment.

Beyond per-example classification latency, we also evaluate POLICYGUARD-4B in a closed-loop web-agent simulation, where it acts as an online interceptor before agent actions are executed. Results in Appendix C show that POLICYGUARD-4B improves Completion-Under-Policy from 10.20% to 16.33% with negligible cumulative latency overhead.

## 5. Conclusion

In this work, we introduced POLICYGUARDBENCH, the first large-scale benchmark for detecting policy–trajectory violations, and proposed POLICYGUARD-4B, a lightweight guardrail model that achieves strong accuracy, cross-domain generalization, and efficient inference. Our experiments demonstrate that safety-oriented guardrails fail to transfer to compliance detection, while policy-specific guardrails can effectively anticipate violations even from early prefixes. These results highlight policy compliance as a distinct and critical dimension of agent reliability, and show that accurate, efficient, and generalizable guardrails are feasible at small scales—laying the foundation for future research on trustworthy policy-compliant agents.

## Impact Statement

This paper introduces POLICYGUARDBENCH and POLICYGUARD-4B, which aim to improve the reliability and safety of autonomous web agents by detecting policy violations. Our work contributes to the broader goal of developing trustworthy AI systems that can operate under strict human-specified constraints. By providing a benchmark and efficient guardrail models, we hope to facilitate research into preventing unintended or non-compliant agent behaviors in real-world deployments. While our guardrail model improves safety, we acknowledge that no detection system is perfect; reliance on such models should be accompanied by other safety layers and human oversight in high-stakes environments. There are no other specific negative societal consequences we feel must be highlighted here.

## Limitations

POLICYGUARDBENCH represents web-agent trajectories as text-based action sequences. While this abstraction enables scalable policy–trajectory matching and efficient guardrail training, it may omit visual or interface-level context needed for realistic GUI reasoning, such as layout information, screenshots, or action grounding. Some failures in our error analysis arise from these abstraction gaps. In addition, although our leave-one-domain-out evaluation shows strong transfer across WebArena-style domains, broader validation on coding agents, embodied agents, and enterprise workflows remains an important direction for future work.

## Acknowledgement

This work was partially supported by a gift fund from Uniphore.

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

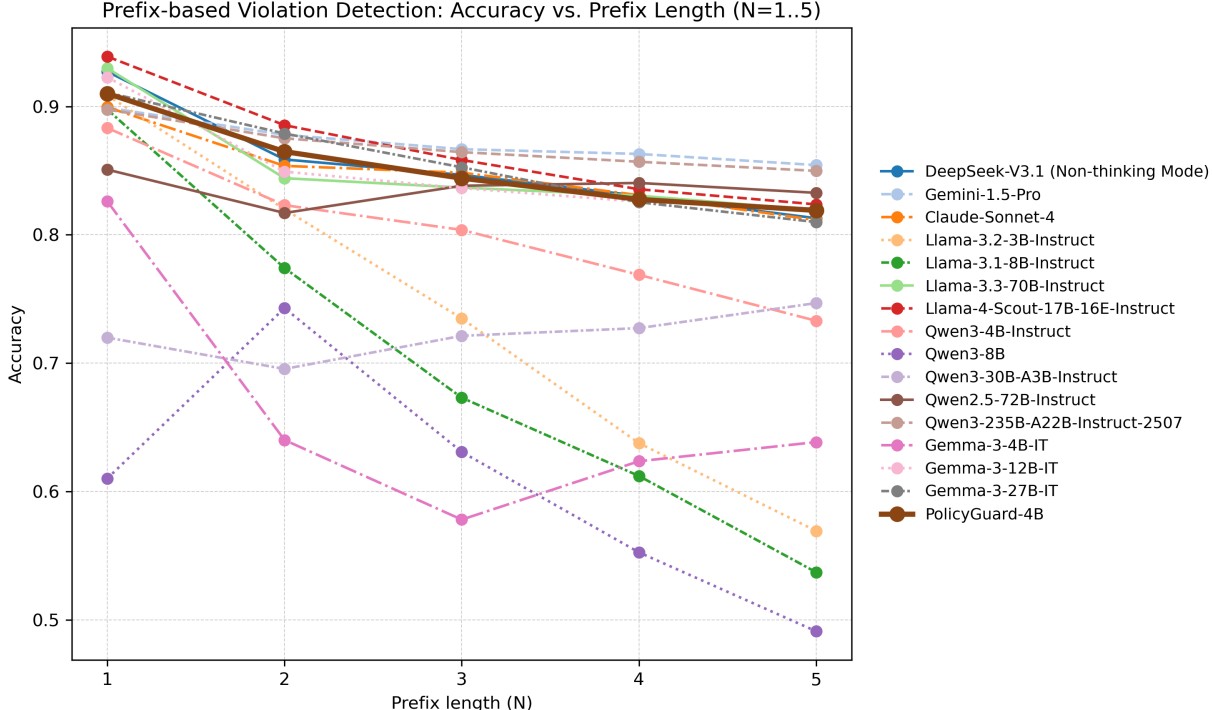

*Figure 3.* Prefix-based violation detection accuracy across *all evaluated models*. Accuracy is generally highest at $N = 1$ and decreases as prefix length increases.

## A. Optimization Details

We summarize the hyperparameters and optimization setup used in our experiments in Table 6.

*Table 6.* Hyperparameters and optimization details for model training.

| Hyperparameter | Value |
|---|---|
| Learning rate | $1 \times 10^{-5}$ |
| Train batch size (per device) | 2 |
| Eval batch size (per device) | 8 |
| Seed | 42 |
| Distributed training | 4 devices |
| Gradient accumulation steps | 8 |
| Total effective train batch size | 64 |
| Total effective eval batch size | 32 |
| Optimizer | AdamW |
| Learning rate scheduler | Cosine |
| Warmup ratio | 0.1 |
| Training epochs | 3 |

## B. Prefix-based Trends Across Model Scales

To complement the main results, Figure 3, Figure 4, and Figure 5 illustrate prefix-based violation detection accuracy broken down by model scale.

Figure 3 provides an overview across all evaluated models, showing the general trend that performance is highest at $N = 1$ and decreases as prefixes become longer. To further clarify scale-dependent behavior, we split models into two groups:

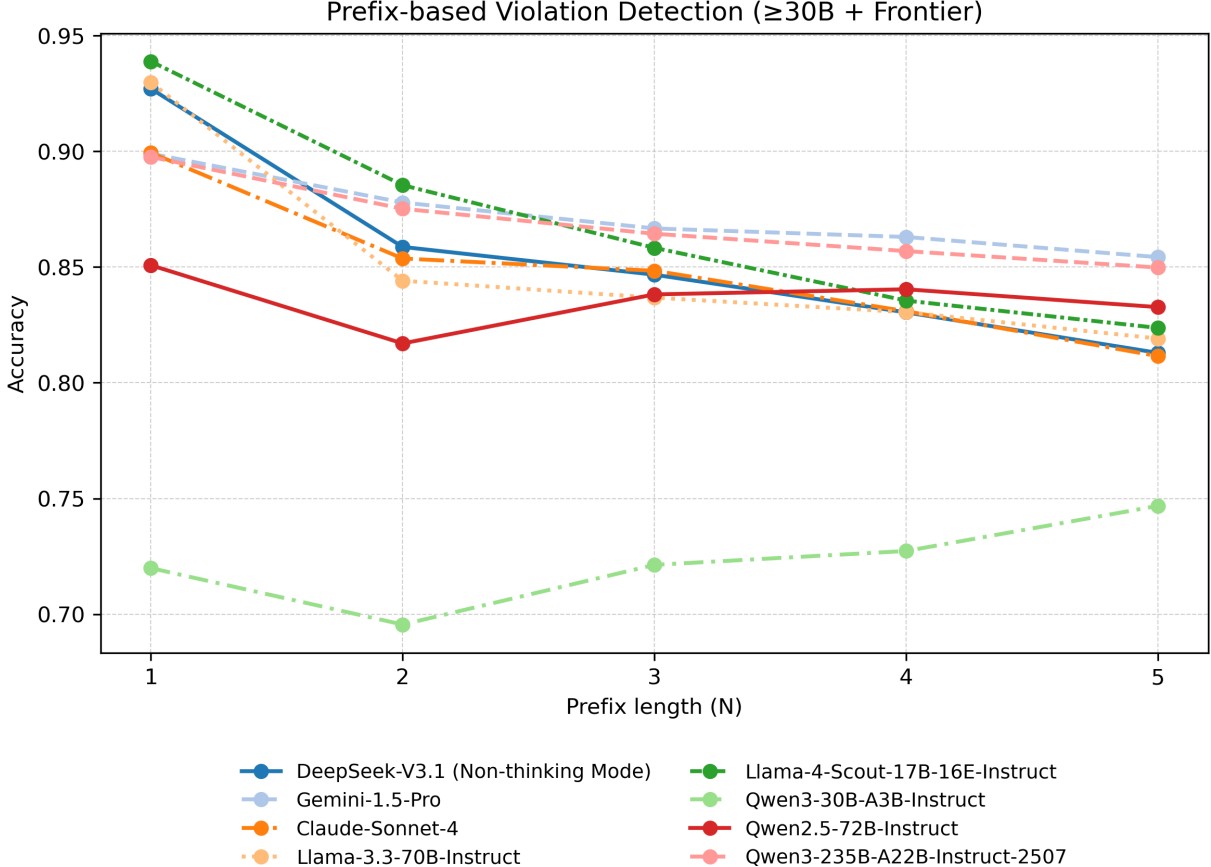

*Figure 4.* Prefix-based violation detection accuracy for *large-scale models* ($\geq$30B parameters and frontier systems). Performance is strong at $N = 1$ and declines moderately as prefix length increases.

large-scale and frontier models ($\geq$30B parameters), and smaller open-source models (<30B, including POLICYGUARD-4B).

Large-scale and frontier models, as shown in Figure 4, start with strong performance at $N = 1$, and although accuracy decreases as more actions are observed, the degradation is relatively moderate. This indicates that larger models can preserve robustness under longer prefixes, albeit at high inference cost.

In contrast, smaller open-source models, as shown in Figure 5, exhibit greater variance and sharper accuracy drops as $N$ increases. Notably, POLICYGUARD-4B maintains stable accuracy across prefix lengths and remains competitive with much larger systems, underscoring its efficiency–robustness advantage. These trends are consistent with the averages reported in Table 3, but reveal more fine-grained behavior across prefix lengths.

## C. Closed-loop Web-agent Simulation

To assess whether POLICYGUARD-4B improves downstream agent behavior beyond offline classification, we evaluate it in a closed-loop web-agent simulation. We use POLICYGUARD-4B as an online interceptor that checks proposed agent actions against the active policy before execution. When a potential violation is detected, the guardrail blocks the action and triggers a retry.

Without intervention, the baseline agent achieves a Completion-Under-Policy (CuP) of only 10.20%, as it frequently proposes actions that violate policy constraints. With POLICYGUARD-4B deployed as the online interceptor, CuP improves to 16.33%, corresponding to a 60% relative improvement. The guardrail produces a 71.97% block rate on violating proposals, indicating that it can prevent many non-compliant actions before execution.

We further measure the runtime overhead of POLICYGUARD-4B within the complete closed-loop session on an H100 GPU.

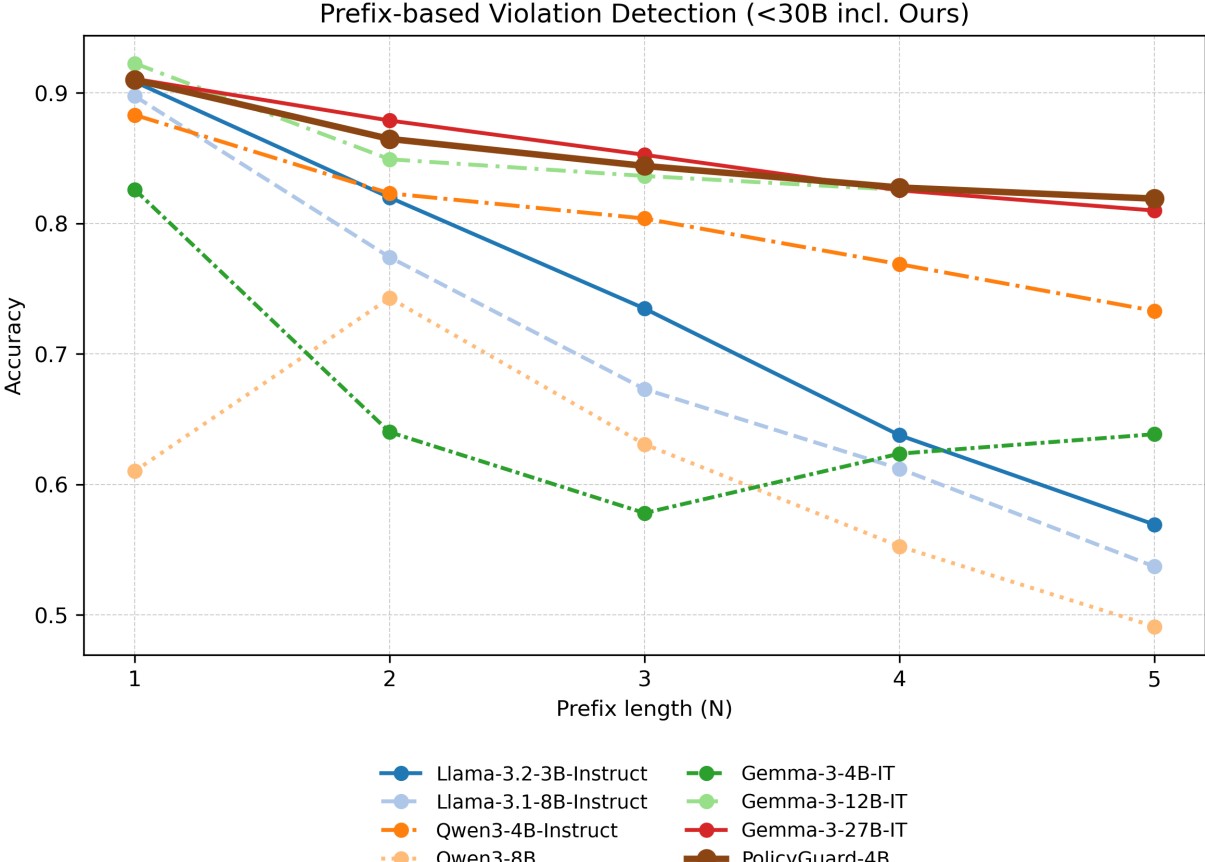

*Figure 5.* Prefix-based violation detection accuracy for *smaller open-source models* (<30B, including POLICYGUARD-4B). Smaller models show greater variance and sharper drops in accuracy, whereas our lightweight POLICYGUARD-4B achieves robust and competitive results across prefix lengths.

*Table 7.* Fine-grained error analysis of POLICYGUARD-4B on the main test set.

| Category | Result |
|---|---|
| Overall errors | 1,091 / 12,000 |
| Error type | 90.3% false negatives, 9.7% false positives |
| Trajectory length | 10.0% error on long ($> 9$ steps), 8.4% on short ($\leq 9$ steps) |
| Domain breakdown | Shopping 10.2%, Reddit 9.3%, GitLab 8.9%, Map 6.0% |
| Primary failure modes | Conditional scroll rules, textbox/action ambiguity, missing final confirmations |

The average inference latency is 26.8 ms per guardrail call. For a typical 10-step long-horizon task, the cumulative guardrail overhead is approximately 0.27 seconds. Since the backbone agent requires 2.86 seconds on average to complete each action, this overhead is negligible relative to the agent execution time.

## D. Fine-grained Error Analysis

To better understand the remaining failure modes of POLICYGUARD-4B, we conduct a fine-grained analysis over the 1,091 misclassified examples in the main test set. The errors are highly asymmetric: 90.3% are false negatives and 9.7% are false positives, indicating that the model is more likely to miss subtle violations than to over-block benign actions.

Errors are also correlated with trajectory and environment complexity. The error rate is higher on long trajectories with more than 9 steps than on shorter trajectories, suggesting that cumulative constraints become harder to track as more actions are

composed. Across domains, most errors arise in Shopping, Reddit, and GitLab, while Map produces relatively few errors. A manual inspection of 100 sampled errors shows that common failure modes include missed conditional scroll prohibitions, confusing textbox clicks with text entry, and missing final-step confirmations. These cases suggest that the main limitation is not generic binary classification, but fine-grained grounding of textual action abstractions to interface-level semantics.

