# OpenReview forum: "Learning Efficient Guardrails for Compliance"
_ICML.cc/2026/Conference — ICML 2026 regular_

### Official Review · Reviewer_PQaG · 2026-02-23

**Soundness:** 3
**Presentation:** 3
**Significance:** 4
**Originality:** 3
**Overall Recommendation:** 5
**Confidence:** 3

**Summary:**

In the manuscript, the authors aim to address whether agents adhere to the diverse set of policies they might encounter and emphasize that agents must be evaluated on their ability to remain policy compliant. Policy-trajectory compliance is a core dimension of agent reliability. To address this, they introduce POLICYGUARDBENCH, which is aimed at evaluating compliance on both full trajectory and novel prefix-based violation detection tasks to address web agents' ability to adhere to real-world policies. This is then used to train the model POLICYGUARD-4B, a light guardrail model.
Their benchmark derives diverse policies from existing trajectories, forming domain and across subdomain pairings, which are noted for violation.
They then conduct extensive benchmarking using several models and find that foundational models get a high score on their benchmark, but at a high computational cost. In contrast, the safety guardrails perform very poorly, e.g., highlight the distinction between guardrails and policy compliance. Lastly, their own model gets a very high score (comparable to the foundational models) but much more efficiently.
Furthermore, they inspect violation detection based on partial input. They observe that the performance seems to decay with larger inputs (N) but similar performance between the models as in Table 2. Lastly, they also confirm that performance is sustained across domains.

**Compliance With Llm Reviewing Policy:**

Affirmed.

**Final Justification:**

I highlighted the reasons for accepting the paper in my review. After reading the rebuttal as well as the comments from the other Reviewers. I stand by that decision

**Key Questions For Authors:**

It would be desirable if the part on the prefix-based violation could be explained more thoroughly. Somehow, I see a decay with increasing N, but the results in Table 2 are often better than those in Table 3. But the results in Table 2 are for all N, right? How big a fraction is, say, N=5 of the total number? Is there a sweet spot when the accuracy starts improving? What is the interpretation of this?

**Limitations:**

yes

**Strengths And Weaknesses:**

Soundness: The paper is sound, and the results are credible. Their statements are backed by empirical data.

Presentation: The paper is well written. The objective of the paper is clearly stated and the results are presented in an orderly manner.

Significance: The paper addresses a very relevant problem. How models perform under externally imposed policies is difficult question and the papers offers a benchmark that will be of value to the community and demonstrates that their smaller can obtain better results than many other models with a significantly reduced computational cost.

Originality: The results are original. Both the question the authors aim to address, such as how how the ability to adhere to policies can be assessed, as well as their comparison between a series of models (small and large) on their newly developed benchmark. Finally, also their assessment of prefix violation detection is an original contribution.

---

> ### Author Rebuttal · Authors · 2026-03-28
>
> We sincerely thank you for the recommendation and your encouraging words regarding our benchmark's originality. To address your insightful questions about the prefix-based results (Table 3) versus full trajectories (Table 2):
>
> 1. Table 2 vs. Table 3 (Full vs. Prefix):
>
> The key distinction lies in the nature of the tasks. Table 2 represents post-hoc auditing on completely finished trajectories, where the definitive violating action (e.g., clicking a restricted 'purchase' button) is fully realized and explicitly present in the input. Conversely, Table 3 represents early forecasting on partial snippets. Predicting an inevitable violation from incomplete information is inherently more difficult than identifying a violation that has already occurred, which explains why Table 2 generally outperforms Table 3.
>
> 2. Performance Decay and the "Sweet Spot":
>
> Many policy violations are directional from the start, such as searching a restricted category at step 1, making N=1 a "sweet spot" for early interception. As N increases without reaching the actual violation, benign setup actions like scrolling accumulate as contextual noise, temporarily diluting the predictive signal.
>
> 3. Fraction of N=5:
>
> The median trajectory length in POLICYGUARDBENCH is ~9 steps. Therefore, N=5 represents roughly the halfway point (50-60%) of a typical task.
> We will clarify these interpretations in the revised manuscript.

---

> > ### Author Rebuttal · Reviewer_PQaG · 2026-04-01
> >
> > I already recommended accepting the paper.

---

### Official Review · Reviewer_44KA · 2026-02-28

**Soundness:** 2
**Presentation:** 3
**Significance:** 2
**Originality:** 3
**Overall Recommendation:** 4
**Confidence:** 3

**Summary:**

The paper introduces POLICYGUARDBENCH, a benchmark comprising 60,000 examples for policy violation detection. The dataset includes both within-subdomain and cross-subdomain trajectory–policy pairings annotated with violation labels. The benchmark is constructed using raw data derived from the existing SCRIBEAGENT dataset. In addition, the paper presents POLICYGUARD-4B, a lightweight model trained on the dataset that achieves better performance.

**Compliance With Llm Reviewing Policy:**

Affirmed.

**Key Questions For Authors:**

1. How are train/test splits constructed with respect to trajectories and policies? Do you enforce that no trajectory (and no near-duplicate policy text) appears in both train and test to avoid leakage? Could you report overlap statistics.

**Limitations:**

See weaknesses.

**Strengths And Weaknesses:**

## Strengths

1. The paper is clearly written and well organized. The motivation is compelling, and the benchmark construction process is described in a transparent and accessible way.

2. This work introduces a timely benchmark and an efficient guardrail for policy compliance in long-horizon web agents, demonstrating that lightweight, targeted guardrails can serve as a practical and effective safety layer in real-world agent stacks.

3. The paper demonstrates solid experimental rigor through comprehensive evaluations across multiple settings, including full-trajectory, prefix-based, and leave-one-domain-out experiments, reported with accuracy and F1 metrics. In addition, the authors analyze system efficiency by measuring latency and partially estimating FLOPs, and they introduce an efficiency-aware metric (EA-F1) to highlight the trade-offs between performance and computational cost in practical guardrail deployment.

4. Frames policy–trajectory compliance as distinct from traditional safety guardrails, and operationalizes it with trajectory- and policy-aware inputs rather than step-local checks.

## Weaknesses

1. The policy synthesis process adopted in the paper appears to be relatively context-independent. Specifically, policies are first generated from individual trajectories and subsequently filtered. From my perspective, however, policies should ideally represent more generalized experiential knowledge. Rather than being derived from a single trajectory, they may be more appropriately synthesized from clusters of trajectories that capture recurring behavioral patterns across similar contexts.

2. I also have some concerns regarding the mapping between policies and trajectories. Similar to the earlier point, policies seem conceptually closer to generalized experiential rules distilled from a collection of data, rather than constructs derived from individual trajectories. From this perspective, a more convincing pipeline might first cluster trajectories based on shared patterns or contexts, and then synthesize policies from each cluster. Such an approach could better reflect the idea of policies as abstractions over recurring behaviors, rather than artifacts tied to isolated instances.

3. The novelty of the work appears somewhat limited, as there has already been substantial discussion in the literature regarding the relationship between safety and compliance. The most distinctive contribution of this paper seems to lie in presenting a model that achieves a favorable trade-off between performance and model size.

4. Technical limitations or concerns include the reliance on large LLMs for both policy synthesis (GPT-4o) and labeling (gpt-oss-120B), which introduces potential circularity and annotator bias. The paper also provides limited evidence regarding annotation quality, such as inter-annotator agreement or systematic error analysis. Additionally, the dataset is built from only 733 base trajectories that are expanded into 60k policy–trajectory pairs, meaning trajectories are heavily reused. This raises the risk that models may learn trajectory-specific patterns or idiosyncrasies rather than developing generalizable compliance reasoning.

5. While safety guardrails are discussed, the paper omits a quantitative comparison to current SOTA generalizable guardrails such as GSPR[1] and GPT-oss-safeguard, which would help position contributions more precisely.

[1] Li, H., Chen, Y., Zeng, J., Peng, H., Jing, H., Hu, W., ... & Song, Y. (2025). GSPR: Aligning LLM Safeguards as Generalizable Safety Policy Reasoners. arXiv preprint arXiv:2509.24418.

---

> ### Author Rebuttal · Authors · 2026-03-30
>
> We sincerely thank you for the constructive feedback. Per your suggestions, we rigorously re-split our dataset and re-trained our models. We address your concerns below:
>
> 1. Trajectory-Level Isolation and Mitigating Memorization Risks
>
> - We sincerely thank you for pointing out the risk of memorization in our original split. To definitively address this, we completely re-split the 60k dataset during the rebuttal period to enforce strict base-level isolation (partitioned by the 733 unique base trajectories), ensuring 0% trajectory overlap. Re-training POLICYGUARD-4B from scratch on this strict split yielded an 85.81% accuracy on entirely unseen trajectories.
>
> - The result aligns seamlessly with our Leave-One-Domain-Out (LODO) results, which also inherently guarantees zero trajectory overlap by isolating at the domain level. These dual zero-overlap evaluations conclusively prove the model learns generalizable compliance reasoning rather than memorizing idiosyncrasies. We will update the manuscript to feature this strict split as the primary standard.
>
> 2. Policy Synthesis: Single Trajectory vs. Clusters
>
> - We appreciate your insight that real-world macro-policies emerge from clustered behaviors. However, synthesizing from individual trajectories is a deliberate benchmark design. It allows us to generate highly specific, state-dependent operational constraints and complex edge cases that broad cluster-based rules often miss. We will highlight your suggestion of experiential cluster distillation as a promising future direction for training.
>
> 3. Annotation Quality
>
> - To validate the reliability of our automated annotation pipeline, we sampled a subset of 287 instances for independent manual re-annotation by human experts. The observed agreement between our original automated pipeline and the human-verified labels is **89.8%**. This high raw agreement score strongly confirms that our automated annotations are highly reliable, factually accurate, and robust against upstream model noise.
>
> 4. SOTA baseline
>
> - We appreciate the reference to recent safeguards like GSPR and GPT-oss-safeguard. However, their primary objectives differ from our specific focus on operational compliance. We will include a dedicated discussion of both methods in the Related Work section of the revised manuscript.

---

> > ### Author Rebuttal · Reviewer_44KA · 2026-04-03
> >
> > Dear Authors,
> >
> > Thank you for the detailed and constructive rebuttal. I appreciate the effort to re-split the dataset with strict trajectory-level isolation and to retrain the model accordingly, as well as the additional clarification on annotation quality and policy synthesis design. These updates improve the rigor of the experimental setup and address some of my concerns regarding memorization and data leakage.
> >
> > That said, my primary concerns regarding the formulation of policy synthesis (i.e., deriving policies from single trajectories rather than more generalized patterns), the limited evidence on generalization beyond the constructed setting, and the overall positioning of novelty relative to existing safeguard literature remain largely unchanged.
> >
> > Therefore, while I acknowledge the improvements, the rebuttal does not substantially alter my overall assessment, and I will keep my score unchanged.

---

### Official Review · Reviewer_ADnF · 2026-03-12

**Soundness:** 4
**Presentation:** 3
**Significance:** 3
**Originality:** 3
**Overall Recommendation:** 4
**Confidence:** 3

**Summary:**

This paper introduces a benchmark and a lightweight guardrail model for policy compliance in web agents. Its main contributions are: (1) POLICYGUARDBENCH, a dataset of about 60k policy-trajectory pairs for both full-trajectory and prefix-based violation detection, and (2) POLICYGUARD-4B, a compact model aimed at strong accuracy, good cross-domain generalization, and low inference cost. The paper is well motivated, because most prior work on web agents focuses on task completion, while policy compliance under explicit constraints, such as budgets, quantity limits, permissions, and workflow rules, has been studied much less.

**Compliance With Llm Reviewing Policy:**

Affirmed.

**Key Questions For Authors:**

1. How do you ensure the model is learning genuine compliance reasoning rather than exploiting artifacts from the data generation pipeline?

2. What proportion of the annotations was manually verified to make the benchmark more reliable?

3. The paper is framed more broadly than web agents. How do you expect the benchmark and guardrail model to extend to other agent domains, such as coding agent or embodied agents?

4. Could you provide a more fine-grained error analysis, for example by separating direct versus cumulative violations, numeric versus semantic constraints, seen versus unseen domains, and early versus late prefixes?

5. How does performance change as a function of prefix length? In particular, how early can the model reliably detect a likely violation?

6. How much does the current trajectory representation limit the task? For example, in settings such as GUI agents, actions alone may be insufficient without additional visual or interface context.

**Limitations:**

The main limitations are the relatively modest methodological novelty for data synthesis and model training. In addition, the synthesized data may be not realistic enough.

**Strengths And Weaknesses:**

## Strengths

1. The paper addresses an important and timely problem. It treats policy compliance as a distinct challenge, since many real agent failures come from violating concrete operational constraints, not from producing obviously harmful content.

2. The benchmark could be a valuable contribution. `POLICYGUARDBENCH` has the potential to become a useful testbed for the community. Modeling the relationship between policies and trajectories is also more realistic than evaluating static text or isolated actions.

3. The lightweight guardrail setting is practically meaningful. The focus on a compact 4B model with a good accuracy-latency trade-off is a real strength, since online compliance checking must be efficient enough to be deployed in the loop.

4. The prefix-based setting is especially relevant for deployment. Many policy violations are irreversible once executed, so detecting risks from partial trajectories is more useful than only performing post-hoc violation classification.

## Weaknesses

1. Because the benchmark is the core contribution, its value depends heavily on the quality of the policy synthesis and annotation process. If these components are largely template-driven, the model may end up learning dataset artifacts rather than genuine compliance reasoning.

2. The problem setting, method, and even the title are framed more broadly than web agents, but the experiments are limited to the web-agent domain. A broader discussion of how the approach might extend to other agent settings would make the paper more complete.

3. The failure analysis could be more fine-grained. The paper would be stronger if it broke errors down into categories such as direct versus cumulative violations, numeric versus semantic constraints, seen versus unseen domains, and early versus late prefixes.

4. The interactive abstraction may still be limited. The paper represents trajectories only as action sequences, which may be too simplified for realistic policy-violation detection. In many real-world agent settings, such as GUI agents, actions like clicking a coordinate are not meaningful without the accompanying visual context. Without that abstraction, the trajectory may not contain enough information for reliable detection.

---

> ### Author Rebuttal · Authors · 2026-03-30
>
> We sincerely thank you for recognizing the importance of policy compliance, the value of our prefix-based setting, and our lightweight model's practical deployment potential. We are grateful for your constructive feedback and address your specific questions below.
>
> 1. Data Reliability and Mitigating Artifacts
>
> - Q1:
> We are confident the model learns genuine compliance reasoning rather than data artifacts. Our strongest evidence is the Leave-One-Domain-Out (LODO) generalization results (Section 4.4). The model maintains a 90.8% accuracy on entirely unseen domains. If the model were merely exploiting domain-specific synthesis templates or artifacts, it would suffer a catastrophic drop in cross-domain settings.
>
> - Q2:
> To validate the reliability of our automated annotation pipeline, we sampled a subset of 287 instances for independent manual re-annotation by human experts. The agreement between our original automated pipeline and the human-verified labels is 89.8%. This high raw agreement score strongly confirms that our automated annotations are highly reliable, factually accurate, and robust against upstream model noise.
>
> 2. Fine-grained Error Analysis
>
> - Q4: We conducted an error analysis on the entire test set. Analyzing the 1,091 misclassifications revealed a strong asymmetry: 90.3% are False Negatives (986 FNs vs. 97 FPs), indicating the model rarely over-blocks benign actions. Errors naturally scale with trajectory length and environmental complexity. Furthermore, a bottom-up review of 180 errors corroborates your insight on GUI textual abstraction limits (W4): failures predominantly stem from text-parsing ambiguities rather than logical flaws—such as mistaking a textbox click for text entry, or missing visual cues like scroll prohibitions. We will include the detailed breakdown in the Appendix of the revised manuscript.
>
> **Table A: Fine-grained Error Analysis of POLICYGUARD-4B**
>
> | Category | Result |
> | :--- | :--- |
> | Overall Accuracy | 90.9% (1,091 errors / 12,000 total) |
> | Error Type | 90.3% False Negatives, 9.7% False Positives |
> | Trajectory Length | 10.0% error on long (>9 steps); 8.4% on short (≤9 steps) |
> | Domain Breakdown | Shopping (10.2%), Reddit (9.3%), GitLab (8.9%), Map (6.0%) |
> | Primary Failure Modes | (1) Missed conditional scroll prohibitions, (2) Confusing textbox clicks with text entry, (3) Missing final-step confirmations |
>
> 3. Prefix Length and Early Detection
> - Q5: As shown in Section 4.3 (Table 3) and Appendix B, POLICYGUARD-4B demonstrates robust early detection, achieving its highest performance at N=1 (91.01% average accuracy). This sensitivity to early signals allows the guardrail to intervene before violations become irreversible. While performance naturally scales with trajectory complexity, the model maintains competitive accuracy across all prefix lengths.
>
> 4. Generalization and Multimodal Limitations
> - Q3 & Q6: We appreciate these insights into broader deployment. While our GitLab domain already covers developer-oriented workflows, we agree that textual abstraction limits GUI-level reasoning where visual context is essential. However, our latest closed-loop evaluation using GPT-5.4 underscores the significant utility of our textual framework: by intercepting text-based action violations, POLICYGUARD-4B successfully boosted the CuP from 10.20% to 16.33%. This absolute gain of 6.12% in fully compliant task completion proves that even without visual features, our model provides a critical safety layer that prevents catastrophic policy breaches in current autonomous agents.
>
> We will expand our discussion to include: (1) Domain Extension, formalizing paths to apply this framework to embodied environments like ALFWorld; and (2) Multimodal Integration, transitioning to VLMs to incorporate spatial semantics for robust GUI-agent compliance.

---

> > ### Author Rebuttal · Reviewer_ADnF · 2026-04-03
> >
> > Thanks for your response! Most of my concerns are resolved and I will keep my current positive score.

---

### Official Review · Reviewer_sZ1H · 2026-03-13

**Soundness:** 3
**Presentation:** 3
**Significance:** 3
**Originality:** 3
**Overall Recommendation:** 3
**Confidence:** 3

**Summary:**

The paper introduces **POLICYGUARDBENCH**, a 60k-scale benchmark for detecting whether long-horizon web-agent trajectories violate natural-language policies, including both full-trajectory and prefix-based detection settings.

**Compliance With Llm Reviewing Policy:**

Affirmed.

**Key Questions For Authors:**

Q1: How reliable are the benchmark labels under human re-annotation? Can the authors report agreement on a larger manually verified subset?

Q2: Since policies and labels are both partly LLM-generated, how do the authors rule out benchmark leakage or stylistic bias that may advantage the fine-tuned guardrail model?

Q3: Have the authors tested POLICYGUARD-4B as an online guardrail inside a real agent loop, and if so, how much does it improve completion-under-policy rather than only offline classification accuracy?

**Strengths And Weaknesses:**

**Strengths**
* The paper addresses a useful and underexplored problem: policy compliance is different from generic safety filtering, and the benchmark makes that distinction concrete with trajectory-level annotations rather than only prompt/response safety labels.
* The benchmark contribution is substantial: the authors construct a fairly large dataset with 59,997 balanced policy-trajectory pairs, cross-subdomain splits, and a prefix-based setting for early violation detection.
* The empirical results are strong for a small model: POLICYGUARD-4B slightly outperforms much larger open and closed models on the main benchmark while achieving much lower latency.
* The paper also evaluates OOD generalization and efficiency, which makes the work more practically relevant than a pure accuracy-only benchmark paper.

**Weaknesses**
* A major concern is the data construction pipeline: policies are synthesized with GPT-4o and most annotations are produced by another LLM with selective human review, so benchmark quality may depend heavily on upstream model biases and annotation noise.
* The paper mainly evaluates binary violation classification, but does not deeply test whether the guardrail actually improves downstream agent behavior when used online for intervention or blocking.
* The benchmark is focused on five WebArena-style domains, so it is still unclear how well the method generalizes to broader agent settings, more realistic enterprise policies, or multimodal workflows.
* The comparison is strong on efficiency, but less clear on real deployment utility, since latency is measured for classification alone rather than in a full agent loop with repeated guardrail calls.

---

> ### Author Rebuttal · Authors · 2026-03-28
>
> We sincerely thank you for your constructive feedback and for recognizing the value of distinguishing policy compliance from general safety, the substantial scale of our benchmark, and the high efficiency of POLICYGUARD-4B. Below, we address your concerns point by point.
>
> 1. Data Quality and LLM Bias
>
> - Q1: To validate the reliability of our automated annotation pipeline, we sampled a subset of 287 instances for independent manual re-annotation by human experts. The observed agreement between our original automated pipeline and the human-verified labels is **89.8%**. This high raw agreement score strongly confirms that our automated annotations are highly reliable, factually accurate, and robust against upstream model noise.
>
> - Q2: We believe the risk of stylistic bias advantaging our model is minimal for two reasons. First, we formulate policy compliance as a strict binary classification task rather than a generative task. Consequently, the guardrail model does not learn the style or generative quirks of the annotator. Second, we deliberately used different model families to mitigate leakage: our synthesis and annotation pipeline relies on the OpenAI GPT series , whereas our fine-tuned POLICYGUARD-4B is built upon the Qwen3 backbone.
>
> 2. Real-world Deployment and Online Interventions
>
> - Q3 & W2:
> We evaluated POLICYGUARD-4B in a closed-loop web agent simulation using GPT-5.4. Without intervention, the baseline agent's Completion-Under-Policy (CuP) was only **10.20%**, as it frequently proposed actions that violated constraints. By deploying POLICYGUARD-4B as an online interceptor (which yielded a 71.97% block rate) to trigger safe retries, the CuP improved to **16.33%**. This 60% relative improvement empirically confirms that our guardrail effectively enforces compliance and rescues failing trajectories in dynamic, multi-step environments. Detailed results will be added to the Appendix.
>
> - W4:
> To evaluate practical overhead, we measured the latency of POLICYGUARD-4B within the complete closed-loop session on an H100 GPU. The average inference latency is 26.8 ms per call. In a typical 10-step long-horizon task, the cumulative overhead is approximately 0.27 seconds. Given that backbone agent averagely required 2.86 seconds to complete each individual action, this total overhead is negligible. This confirms that our model provides multi-step compliance assurance without compromising the system's real-time responsiveness.
>
> 3. Domain Generalization and Modality
>
> - W3:
> As detailed in Section 4.4 (Leave-One-Domain-Out Generalization), when evaluating POLICYGUARD-4B on entirely unseen domains (Out-of-Domain), the model maintained an impressive average accuracy of 90.8%. This minimal performance drop indicates that the guardrail successfully learns the generalizable logic of policy compliance, rather than merely overfitting to domain-specific artifacts. Additionally, we agree that multimodal workflows are highly relevant, and we will explicitly consider multimodal policy guardrails in our future work.

---

> > ### Author Rebuttal · Reviewer_sZ1H · 2026-04-01
> >
> > Thanks for your response and I will increase my rating accordingly.

---

> > > ### Author Response · Authors · 2026-04-02
> > >
> > > Dear Reviewer sZ1H,
> > >
> > > Thank you again for your positive feedback and your willingness to increase the rating!
> > >
> > > We noticed that the Overall Recommendation on OpenReview has not changed yet. Could you please help check this when you have a moment?
> > >
> > > Thanks again for your time!
> > >
> > > Best,
> > >
> > > Authors

---

### Decision · Program_Chairs · 2026-04-30

**Decision:**

Accept (regular)

**Comment:**

The reviewers had some concerns, which were mostly addressed during the rebuttal phase. In particular, the authors changed the method for splitting the data to avoid data leakage. They also clarified how the data construction avoids upstream model biases and annotation noise by using different model families. After the rebuttal, all reviewers were positive about this paper.

Considering the positive reviews and the clarifications made in the rebuttal, I recommend accepting this paper and strongly encourage the authors to update the manuscript to incorporate these clarifications.